# A Novel Multipath Transmission Scheme for Information-Centric Networking

**Yong Xu** [1,2], **Hong Ni** [1,2] **and Xiaoyong Zhu** [1,2,*]

1   National Network New Media Engineering Research Center, Institute of Acoustics, Chinese Academy of Sciences, No. 21, North Fourth Ring Road, Haidian District, Beijing 100190, China
2   School of Electronic, Electrical and Communication Engineering, University of Chinese Academy of Sciences, No. 19(A), Yuquan Road, Shijingshan District, Beijing 100049, China
*   Correspondence: zhuxy@dsp.ac.cn; Tel.: +86-131-2116-8320

**Abstract:** Due to the overload of IP semantics, the traditional TCP/IP network has a number of problems in scalability, mobility, and security. In this context, information-centric networking (ICN) is proposed to solve these problems. To reduce the cost of deployment and smoothly evolve, the ICN architecture needs to be compatible with existing IP infrastructure. However, the rigid underlying IP routing regulation limits the data transmission efficiency of ICN. In this paper, we propose a novel multipath transmission scheme by utilizing the characteristics and functions of ICN to enhance data transmission. The process of multipath transmission can be regarded as a service, and a multipath transmission service ID (MPSID) is assigned. By using the ICN routers bound to the MPSID as relay nodes, multiple parallel paths between the data source and the receiver are constructed. Moreover, we design a path management mechanism, including path selection and path switching. It can determine the initial path based on historical transmission information and switch to other optimal paths according to the congestion degree during transmission. The experimental results show that our proposed method can improve the average throughput and reduce the average flow completion time and the average chunk completion time.

**Keywords:** multipath transmission; path selection; path switching; ICN

## 1. Introduction

According to Cisco, internet users and traffic have grown rapidly over the past five years. By 2023, sixty-six percent of the global population will be internet users and average global fixed broadband speeds will be 110 Mbps [1]. The development of emerging network applications such as virtual reality (VR), ultra-high-definition video, unmanned driving, and industrial automation has put forward new requirements for the existing TCP/IP networks in terms of high bandwidth, massive communication, and low latency. Due to the lack of a native mechanism to support efficient content distribution, the host-centric traditional network architecture has been unable to meet the needs of current network services. With this background, a new network paradigm, information-centric networking (ICN) [2,3], has been proposed to solve the problems faced by existing TCP/IP networks.

Unlike traditional TCP/IP, ICN decouples the locator and identifier of resources, thus avoiding the problems caused by IP semantic overload, such as mobility, security, and scalability [4]. ICN uses a globally unique ID to name network entities and deploys in-network caching, which naturally supports massive content distribution. The development of ICN has led to many related projects. According to the difference in routing and forwarding methods, the existing ICN paradigms can be mainly divided into two types, the ICN paradigm of name-based routing and the ICN paradigm of stand-alone name resolution. The former mainly uses hierarchical and aggregated names, and the name resolution process is coupled with message routing. Typical examples of this paradigm include Content Centric Networking (CCN) [4], Named Data Networking (NDN) [5], and

Content Network (CONET) [6]. However, the deployment of such an ICN paradigm is costly, as significant upgrades to the infrastructure of traditional IP networks are required. The latter mainly uses flat names, and the name resolution process and the message routing process are decoupled, such as MobilityFirst [7], Network of Information (NetInf) [8], and On-Site, Elastic, Autonomous Network (SEANet) [9]. In this type of ICN paradigm, the Name Resolution System (NRS) plays an important role in maintaining the relationship between network address (NA) and identifiers (ID). Publishers of content or service can register their NAs and the corresponding ID with the NRS. Then, users can obtain the NAs of the publishers from the NRS according to the content or service ID, and transmit the data through the NA routing function. Since it is not practical to override the existing IP network, using IP addresses as NAs is a feasible solution [10]. This type of ICN paradigm is more compatible with the existing IP infrastructure, thus enabling smooth evolution. Note that our proposed multipath transmission scheme is based on the latter ICN paradigm.

However, due to compatibility with the existing IP infrastructure, ICN packets are still transmitted along the default "IP path" given by the underlying IP routing regulation. The single path IP routing protocols stubbornly believe that all ICN packets travel from source to destination by the same default shortest path. Therefore, it is difficult to guarantee user demands for high reliability and bandwidth when applied with the IP single path routing protocol, especially in the scenario of massive content distribution in ICN.

In order to solve the problems of the low resource utilization and poor fault recovery capability of single-path methods, some multi-path transmission protocols have been proposed [11,12]. Although Equal-Cost Multipath Routing (ECMP) [11] has been used in data center networks, its improvement effect in asymmetric network topology is not significant. Segment Routing IPv6 (SRv6) [12] has been proposed in recent years to solve the problem of IP routing rigidity, but the excessive header overhead will reduce the data carrying efficiency. In addition, some researchers have focused on how to utilize the characteristics and resources of the underlying IP network at the application layer [13–17]. These overlay methods build relay paths to avoid failure or congestion by detecting information such as delay and packet loss rate of the underlying network in real time. Although the overlay methods complement traditional network protocols, their performance is still unable to break through the bottleneck of the protocol stack due to the deployment at the application layer, and they cannot be directly applied to ICN. Moreover, there have been some attempts to implement multipath forwarding based on multiple addresses of terminal devices identified by globally unique identifiers in the ICN paradigm of stand-alone name resolution [18,19]. However, these methods can only be applied in the scenario of multi-homed terminals.

In this context, we propose a novel multipath transmission scheme applied in an IP-compatible ICN architecture. Our goal is to make full use of the characteristics and functions of ICN and the diversity of IP paths to improve the data transmission efficiency and robustness. The main contents of this paper are as follows:

- We introduce the overall layout of our ICN protocol stack and propose a novel multipath transmission scheme in an IP-compatible ICN architecture. We regard the multipath transmission process as a kind of service and the multipath transmission service ID (MPSID) is assigned. Based on the MPSID, multiple parallel paths can be constructed between data sources and receivers by utilizing ICN routers as relay nodes.
- We propose a path management mechanism to make full use of multipath resources. To reduce the overhead and avoid a poor selection, the initial path can be determined according to the historical transmission information. Besides, by measuring the congestion degree of the selected path during transmission, path switching can be performed to avoid bad paths.
- We conduct a series of experiments to verify the performance of our multipath transmission scheme. The experimental results show that our proposed method has a significant improvement in terms of average throughput, average flow completion time, and average chunk completion time.

The rest of this paper is organized as follows. We review the related research on multipath transmission mechanisms of IP networks and ICN networks, respectively, in Section 2. In Section 3, we describe the overall layout of our ICN protocol stack and the principle and process of ID-based multipath transmission scheme. In Section 4, the path management mechanism is described in detail. Then, we conduct simulation experiments of the multipath transmission scheme and discuss the experimental results in Section 5. Finally, we conclude our research and look forward to future work in Section 6.

## 2. Related Work

Multipath transmission techniques have been extensively studied due to their improvement in transmission robustness and efficiency. Since the multipath transmission scheme we propose is applied in an IP-compatible ICN architecture, we present a comprehensive study on the multipath transmission mechanism of IP networks and ICN, respectively, in this section.

### 2.1. Multipath Transmission Mechanism of IP Networks

According to the difference in design and deployment, the multipath transmission mechanism of the existing IP networks can be classified into the overlay method and the underlay method.

The overlay method creates a virtual overlay topology on the underlying network and forms multiple end-to-end paths through logical links. The Resilient Overlay Network (RON) [13] is a typical example of the overlay method. RON periodically measures the status information of virtual links between overlay nodes, then distributes topology and path information to each neighbor node, and finally forwards traffic by building IP tunnels to avoid congested links. However, the overhead of periodic detection and information distribution is not conducive to large-scale deployment of RON. Based on RONs, some studies have proposed a multi-homing overlay network (MON) [14], which introduces the multi-homing characteristics of devices into overlay routing. Although the MON improves transmission benefits, the high overhead is still its fatal drawback. In order to improve the scalability of the overlay method, the authors in [15] proposed the One-Hop Source Routing (OHSR), which attempts to recover from congestion or failure by randomly selecting optional relay nodes for indirect routing. However, the random selection manner does not guarantee the status of the indirect path. In addition, Path Probing Relay Routing (PPRR) [16] uses a random search method to discover alternative detour paths and determines the path state by probing on demand. When serious congestion or failure occurs on the direct path of the underlying IP network, PPRR can quickly switch to an alternative path. Moreover, the Topology-Aware Reliable Overlay Multipath (TAROM) [17] can accurately find relay nodes based on global topology and routing information, but it is difficult to obtain dynamic internet topology and routing information timely and accurately.

The underlay method uses a layer-2 or layer-3 routing and forwarding mechanism to ensure end-to-end multipath connectivity. ECMP is a classic multi-path routing protocol, which is widely used in Data Center Networking (DCN). ECMP distributes traffic equally over multiple available forwarding ports based on the diversity of the underlying physical paths. However, the traffic is equally distributed, which also determines that ECMP is difficult to apply to asymmetric networks. Therefore, researchers proposed Weighted Cost Multi-Path (WCMP) [20] based on ECMP. WCMP distributes traffic proportionally across paths based on link state. However, WCMP is inflexible and cannot reroute flows based on congestion information. In [21], the authors proposed a dynamic acyclic multipath routing algorithm, which keeps multiple possible next hops and weights for one destination address, and the router assigns packets of the same destination address to multiple next hops according to the estimated weight ratio. Compared with the traditional IP routing protocol, Multi-Protocol Label Switching (MPLS) [22] provides a new network switching method, which maps IP addresses into short and fixed-length labels. MPLS uses label switching instead of IP routing table lookup to significantly improve forwarding efficiency.

As a result of its high price and poor security, MPLS has been stretched under the condition of high bandwidth demand. In addition, researchers have proposed segment routing in the past few years. Segment routing [23] is a packet routing and forwarding mechanism based on source routing. The forwarding path is encoded in the packet header before sending. During the forwarding process, routers can determine who the next hop is according to the header address field. Segment routing has attracted much attention due to its flexible path selection and forwarding. For instance, the authors in [24] proposed an online and offline algorithm for path selection based on segment routing, and the experimental result showed that both methods are effective. SRv6 [12], proposed in recent years, combines the advantages of IPv6 and segment routing, and has been widely deployed on Wide Area Networks (WANs). Although these segment routing technologies can specify the transmission path, the excessive header overhead will reduce the data carrying efficiency. In wireless networks, how to implement self-organizing multipath routing through relay devices has been extensively studied [25,26], but these technologies can only be applied in wireless scenarios.

*2.2. Multipath Transmission Mechanism of ICN*

With the development of ICN, there have been many studies on ICN multipath transmission. In this subsection, we introduce the multipath transmission techniques of the ICN paradigm of name-based routing and stand-alone name resolution, respectively.

In the ICN paradigm of name-based routing, the data packets do not loop since they take the reverse path of request packets, so the multi-path transmission mechanism relies on its flexible forwarding plane in this ICN paradigm. In [27], the authors used three colors to mark the performance of the forwarding port, and the router can preferentially select the port with the best performance to forward the data packet, thus avoiding the congested path. In [28], the authors proposed an on-demand multi-path forwarding mechanism based on the principle of minimum RTT priority, which forwards user requests from different interfaces in proportion according to RTT. In [29], the authors regard the process of request packet forwarding as a multiple attribute decision making problem, and proposed an Entropy-based Probabilistic Forwarding (EPF) strategy. EPF integrates multiple network state metrics to accurately calculate the state of interfaces by objectively assigning weights to the metrics. In [30], the Markov Decision Process (MDP)-based forwarding strategy was introduced, which uses queuing theory to estimate the real-time network state, and builds an MDP model to guide request packets for probabilistic forwarding. In [31], the authors regarded the problem of multi-path congestion control and request forwarding as a global objective optimization problem for maximum throughput and minimum network cost, and derived a set of optimal distributed algorithms for dynamic request forwarding. In addition, some researchers have used reinforcement learning algorithms to design forwarding strategies [32,33]. ICN routers perform probabilistic forwarding to explore potential cache replicas in the network, and then use the learned results to guide request packet forwarding. However, these methods are only suitable for the innovative ICN architecture, so it is difficult to apply them in practice.

In the ICN paradigm of stand-alone name resolution, there have been some attempts to implement multi-path forwarding based on multiple addresses of terminal devices identified by globally unique identifiers. In [18], the authors proposed a novel forwarding scheme in the multi-address scenario in this ICN paradigm. In this scheme, multiple addresses of the device are carried into the destination address group field of the ICN data packet header, which enables each data packet to be matched to multiple interfaces in the hop-by-hop forwarding process, and the router can select an optimal interface for each data packet according to the state information of the interfaces. In [19], the authors proposed a network-assisted multipath transmission mechanism, in which the in-network nodes can split data into different paths according to locally generated policies. Specifically, multiple network addresses can be obtained by querying the NRS according to the globally unique identifier (GUID), so as to establish multiple paths to destination mobile devices.

### 3. Design

We first outline a practical protocol stack of an IP-compatible ICN architecture in this section. Then, we analyze the disadvantages of the single-path transmission method to clarify the motivation of our study. Finally, we describe the principle and process of the proposed multipath transmission scheme in detail.

### 3.1. Protcol Stack

In order to reduce deployment costs, it is reasonable to deploy the new ICN architecture on top of an existing IP network, so as to fully utilize the IP infrastructure. Figure 1 shows a practical ICN protocol stack, in which the ID layer is incrementally deployed on top of the IP layer, thereby extending the functions of the network layer. At the network layer, a protocol called the identifier protocol (IDP) [34] is running, which defines a set of regulations specifying how to operate on the NA according to the ID of the data packet, including adding, deleting, and modifying. Above the ID layer is a transport layer, on which we design a transport protocol [35], including the ICN packet format, transmission process, congestion control mechanism, and retransmission mechanism. The transport layer protocol can provide an efficient and reliable data transmission service, and it belongs to the scope of the transport layer.

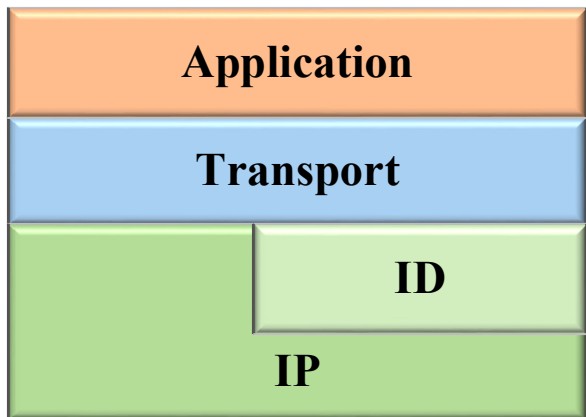

**Figure 1.** The protocol stack of ICN.

In ICN, the chunk is the basic data unit for transmission and caching, which is bound to an addressable globally unique name. Most of the existing switches or routers are two-layer or three-layer structures; therefore, the ICN packet can be normally routed and forwarded on the existing network. In order to achieve smooth evolution, we consider the deployment of hybrid networking, where the entire network consists of IP routers and ICN routers. The ICN router has a complete ICN protocol stack, as well as computing and storage capabilities. In addition, we apply the concept of late binding in ICN [36], which means that the NA of the packets can be modified according to the policy generated by the local computing module during the forwarding process. Here are some typical examples: In the chunk retrieval scenario, the ICN routers can select the optimal replica by modifying the destination NA of the request packets according to the NAs of content providers obtained by querying the NRS via a chunk ID [37]. In mobility scenarios, the mobile device, whose NA is changed, can re-register its NA and ID relationship with the NRS; then, the ICN routers can modify the destination NA of data packets according to the NA obtained by querying the NRS via mobile device ID, so as to ensure transmission continuity [38].

### 3.2. Motivation

However, due to the compatibility with the existing IP infrastructure, ICN packets are still transmitted in a best-effort approach employed by IP routing rules. With the increase

in the number of access devices, the congestion probability of the default path gradually increases. Once congestion or failure occurs, it often takes seconds or even minutes for routing protocols to converge to a normal state [39]. The end-to-end connections may experience outages for seconds or minutes during this process. Therefore, rigid IP single path routing has been unable to meet the demand of massive data distribution. A reasonable data transmission scheme is necessary to improve the transmission reliability and efficiency of such IP-compatible ICN architectures.

With a complete protocol stack, the ICN routers can resolve an ID to network address(es) by initiating a query operation to the NRS. In addition, the ICN routers can also use the IDP to process the network address(es) of packets, which may include adding, deleting, and modifying the network address. Therefore, it is a feasible solution to utilize the ICN routers to act as relay nodes to enhance transmission. In addition, researchers have proven through experiments that the single-hop indirect path formed by one relay node could achieve significant gains in terms of round-trip delay, packet loss rate, and throughput [13]. Based on this finding, an ICN router can form a relay path between the sender and the receiver, so multiple ICN routers can be used for multipath transmission. In the next subsection, we will introduce the design of our proposed multipath transmission scheme in detail.

### 3.3. Overview of Transmission Process

In ICN, network services are treated as a kind of network entity and are labeled with identifiers, such as multicast services [40] and storage services [34]. Inspired by this, we consider multipath transmission as a kind of service and assign the corresponding MPSID. Some ICN in-network routers need to register the relationship of their NAs to the MPSID with the NRS so that they can be addressed by other network devices. These ICN routers are regarded as relay nodes for end-to-end transmission, which can reroute traffic sent by the data source to the destination. Therefore, in addition to the default shortest path, the data source can use multiple suitable relay ICN routers to construct multiple parallel single-hop relay paths between the data source and the receiver.

Figure 2 shows the process of the proposed multipath transmission scheme.

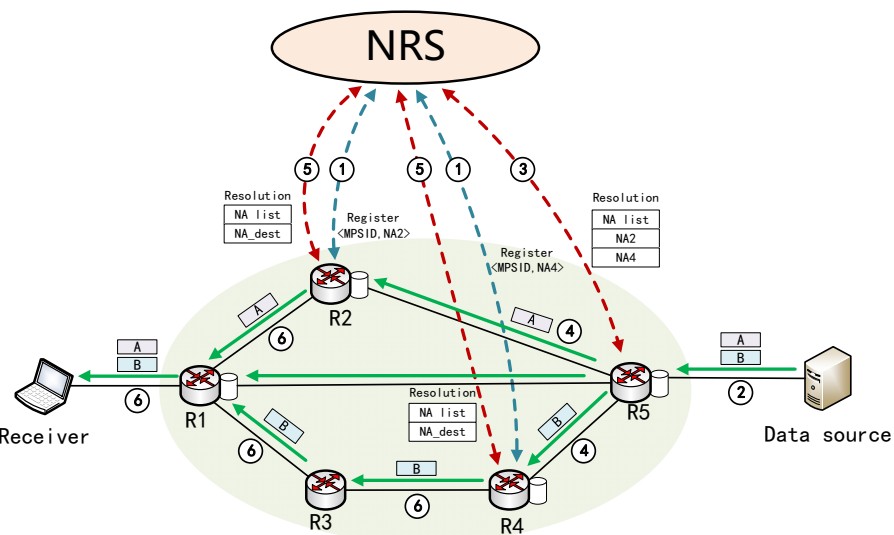

**Figure 2.** An overview of the multipath transmission process.

When the ICN routers (R2 and R4) supporting the multipath transmission service go online, they will register the mapping between its NA and MPSID with the NRS (step 1).

After the connection between the two sides is established, the data source starts transferring local data chunks. Each chunk to be transferred is segmented according to the Maximum Transmission Unit (MTU) and an ICN header is added to form an ICN data

packet. The data source can transmit these packets using the default path; in addition, if a higher bandwidth is required, the data source can mark a multipath transmission label in the preference field of these ICN data packets according to the application requirements, and then send packets to the corresponding edge router (here, R5) (step 2).

For security reasons, the NAs of in-network ICN routers should not be exposed to devices outside the ICN network area, so the query operation to the NRS is initiated by the edge node. When receiving the first marked data packet of the first chunk during this transmission, R5 initiates a query operation to the NRS according to the MPSID to obtain the NA list (here, NA2 and NA4). During the whole end-to-end transmission, the operation of initiating a query to the NRS by the edge router only happens once and the subsequent query operation is not needed (step 3).

According to the node selection strategy, R5 selects several suitable NAs from the NA list and adds them to the selected node set. Then, R5 changes the destination NA of the data packet to one of the selected NAs and caches the network address translation information locally. It is worth noting that the granularity of data scheduling is chunk, so segmented packets of the same chunk will be processed in the same way. Since IPs can be used as NAs in this paper, these data packets will be transmitted to the corresponding relay node (R2 or R4) through an IP routing function (step 4).

When receiving the first data packet of a chunk, the relay ICN routers (R2 or R4) will send a name resolution request to the NRS according to the destination ID to obtain the NA of the receiver device. It should be noted that this query operation only occurs when the first data packet of a chunk is received. The relay router will form NA translation rules locally. When subsequent packets of the same chunk arrive, their destination NAs are automatically changed according the translation rules (step 5).

Finally, the relay ICN router (R2 or R4) modifies the destination NA of the data packet to the NA of the receiver device and forwards it (step 6).

In contrast to some overlay routing methods, our multipath transmission scheme is implemented based on the MPSID at the network layer rather than the application layer. In the process of multipath transmission, three kinds of IDs are mainly involved, which are the IDs of chunks to be transmitted, the ID of the receiver device, and the MPSID. The source ID and destination ID of the data packet are the chunk ID and the receiver device ID, respectively. The MPSID does not appear in the header of the packet, and the edge node can decide whether to use the MPSID according to the preference field. Locators and identifiers are separated, which is one of the characteristics of ICN. In the process of multipath transmission, the MPSID acts as an identifier, and each NA bound to an MPSID acts as a locator, which conforms to the basic characteristics of ICN. In this paper, we refer to the design pattern of flat IDs from MobilityFirst [7], and use a 20-byte flat ID to name network entities, including the MPSID. In addition, the NRS plays a vital role. Firstly, the data source can obtain the NAs of these relay ICN routers by querying the NRS with an MPSID to discover multiple paths. Secondly, the relay ICN routers can query the NRS through the destination ID of the data packets to obtain the NA of the receiver device. Existing ICN multipath transmission mechanisms either rely on its flexible forwarding plane or on the multi-homing feature of devices. To the best of our knowledge, unlike existing mechanisms, we are the first to regard multipath transmission as a service and use an ID to identify it. At the transport layer, we use the previously proposed transport protocol [35], which has congestion control and a retransmission mechanism, to ensure efficient and reliable transmission by adjusting the request sending rate within a single chunk.

## 4. Path Management

In our scheme, an ICN router bound with an MPSID can constitute a one-hop indirect path, which can provide a data relay service for end-to-end transmission. Therefore, making full use of path resources is vital. To this end, we propose in this section a path management mechanism in detail. Path management consists of two phases: path selection and path switching. In the path selection phase, we select several relay nodes as initial temporary

solutions based on historical transmission information. In the path switching phase, we measure the congestion degree of these selected paths. If any of these paths experience heavy congestion, we discard this congested path and find an alternative.

### 4.1. Path Selection

Some studies [41] have shown that only a small number of relay nodes can provide optimal relay paths for most end-to-end transmission pairs, and these nodes usually have high betweenness centrality. Therefore, we make some ICN routers with high betweenness centrality register their NA-MPSID relationship with the NRS. Betweenness centrality [42] of a node $i$ is the sum of the fraction of all-pairs shortest paths that pass through $i$, which is denoted as follows:

$$BC(i) = \sum_{s,t \in V} \frac{\sigma_{st}(i)}{\sigma_{st}} \qquad (1)$$

where $V$ is the set of nodes in topology, $\sigma_{st}$ denotes the number of shortest paths from $s$ to $t$, and $\sigma_{st}(i)$ is the number of shortest paths from $s$ to $t$ that go through $i$.

After obtaining the NA list of the relay nodes by querying the NRS with the MPSID, the edge router needs to select several appropriate NAs from the NA list to form the initial multipath. In [43], the authors point out that relay node selection is an NP-hard problem. Although some node selection algorithms have been proposed in the overlay network [13,15,16], their performance is poor in terms of robustness or scalability. The RON [13] and other similar systems require the relay nodes to probe the entire network and periodically exchange probe information, and are hence not scalable. OHSR [15] adopts a random selection strategy and thus is scalable, but it is hard to avoid poor relay nodes. PPRR [16] can maintain a top set of optional relay nodes for each destination, thus reducing the probing overhead. However, the probe overhead is still not negligible as the number of relay nodes increases. Therefore, we need a more scalable way to discover available relay paths.

To reduce the overhead and avoid a poor selection, we let the edge router maintain an alternate path state table (*PST*). The *PST* records the NAs of the relay nodes and the corresponding transmission success rate (*TSR*). The *TSR* is obtained according to the success rate of a relay node providing data relay service in the past period of time, so it can reflect the relay node's ability to provide multipath transmission service to a certain extent. The *TSR* can be calculated as follows:

$$TSR_i = \frac{s_i}{s_i + f_i} \qquad (2)$$

where $s_i$ and $f_i$ represent the number of times that the path formed by relay node $i$ is congested and not congested, respectively. $s_i$ and $f_i$ are the results of long-term historical observations. If the path formed by the selected relay node $i$ is not congested during one transmission, the corresponding $s_i$ is increased by one; that is,

$$s_i = s_i + 1 \qquad (3)$$

Conversely, if the path is congested during transmission, the corresponding $f_i$ is decreased by 1; that is,

$$f_i = f_i - 1 \qquad (4)$$

In this way, the node's *TSR* can be calculated and updated during every transmission. The larger the *TSR* of a relay node, the higher the probability of it being selected. The rationale for this method is that a relay node that previously provided a sufficiently well-conditioned path to various destinations is likely be selected again with higher probability in the future [15]. There may be some ICN routers registering or deregistering with the MPSID on the NRS; therefore, the edge routers need to update the NA entries of the *PST* according to the NA list of the relay nodes obtained from the NRS. In addition, if a relay

node has not been selected for a long time or a new relay node is added to the *PST*, its *TSR* will be automatically set to the default value. The *PST* has a simple structure and its update does not require additional probe overhead, so it is scalable.

The *PST* can provide prior knowledge, and thus can reduce the difficulty and overhead of node selection. Suppose that *P* is the set of alternative nodes obtained from the NRS and the set of selected nodes is denoted as *S*. Our goal is to pick *k* optimal relay nodes from *P* according to the *PST* and add them to *S* to form initial multiple paths. If all the relay nodes' *TSRs* in the *PST* are the same, the edge routers will randomly select *k*. In this paper, we set the value of k to 3. The reason is that it has been shown in [44] that most of the benefits can be obtained by using three to four relay paths. By selecting more than one relay routers based on the *PST*, the edge router can ensure that a single unlucky selection is not fatal. The path selection strategy based on the *PST* cannot ensure that all selected paths are in a good state, but it can improve the availability of the initial path to a certain extent. The path selection strategy needs to cooperate with the path switching strategy, which will be introduced in the next subsection. The detailed path selection algorithm is shown in Algorithm 1.

---

**Algorithm 1** Path Selection Algorithm

---

1: **Input:** *MPSID, PST, k*
2: **Output:** *S*
3: **Initialization:** *S = NULL, P = NULL*
4: **At the beginning of transmission:**
5:    *P = querybyNRS(MPSID)*
6:    update the NA entries of *PST* according to *P*
7:    *S = SelectTop_K(P, PST, k)*
8: **During transmission:**
9:    execute the path switching algorithm //describe in Algorithm 2
10: **At the end of transmission:**
11:    **for** every $NA_i$ in *S* **do**
12:        $TSR_i = TSR_i + 1$
13:        release *P, S*
14: **end for**

---

*4.2. Path Switching*

As described above, we determine the initial paths based on prior knowledge of the *PST*. However, these ICN relay nodes usually need to carry more traffic and are a frequent location of congestion. During the transmission, once a path experiences heavy congestion, it is necessary to re-examine its availability. To alleviate congestion, a common method is to allocate fewer data chunks to congested paths. This method is feasible when the path is slightly congested for a short time. Once a path is heavily congested for a long time, it means that the path cannot provide high-quality transmission services. In this case, it is wise to delete the congested path from *S* and look for another alternative.

In this paper, we use the packet loss rate to judge whether a sub-path is congested. Considering that a high sampling frequency may cause redundant calculation overhead, we set a time interval, $\Delta t$, during which the receiver samples the instantaneous packet loss rate. According to the experimental results from [45], we set $\Delta t$ to 0.2 s. As shown in Equation (4), the packet loss rate is calculated in a smooth manner to avoid erroneous estimates caused by short-term burst traffic.

$$LOSS_i(t) = \alpha \times loss_i(t) + (1 - \alpha) \times LOSS_i(t - \Delta t) \tag{5}$$

where $LOSS_i(t)$ is the average packet loss rate of path *i*, and $loss_i(t)$ is the ratio of the number of lost packets to the number of transmitted packets during $\Delta t$. The receiver calculates the packet loss rate of each path and feeds the congestion information back to the edge router through request packets, and the latter performs path switching. We

introduce a congestion threshold, $L_{thresh}$. According to the relationship between $LOSS_i(t)$ and $L_{thresh}$, we can judge whether the path is in the heavy congestion state. The path switching algorithm pseudocode is shown in Algorithm 2.

$$congestion\_flag = \begin{cases} 0, 0 < LOSS_i(t) < L_{thresh} \\ 1, L_{thresh} < LOSS_i(t) \end{cases} \quad (6)$$

The main overhead of multipath transmission is determined by the maintenance cost of the path state. The higher the maintenance cost, the higher the complexity. Assume that the number of optional relay nodes is $N$. In a RON, each node needs to maintain the path state with other $N-1$ nodes, so the maintenance cost is O($N^2$). PPRR selects $M$ from $N$ optional nodes to form the top set and maintains the path state between $M$ nodes according to the probing results, so its complexity is O($M^2$). The method adopted in this paper is based on prior information of historical transmission, and does not require any probing operation, so the complexity is also O(1). This shows that our method is scalable. Moreover, we use a practical data scheduling strategy in this paper. To avoid out-of-order arrival of packets of the same chunk, we set the granularity of data scheduling to chunk level instead of packets. The edge node adopts the round robin method based on the packet loss rate, and preferentially allocates the chunks to be transmitted to the path with the smallest packet loss rate.

---

**Algorithm 2** Path Switcing Algorithm

---

1: **Input:** *loss(t)*, *LOSS(t-Δt)*, *PST, P*
2: **Output:** *S*
3: **for** every Δt during transmission **do**
4:     count the packet loss rate $loss_i(t)$ of selected path
5:     $LOSS_i(t) = α × loss_i(t) + (1-α) × LOSS_i(t-Δt)$
6:     **for** every $NA_i$ in $S$ **do**
7:       **if** $LOSS_i(t) > L_{thresh}$ **then**
8:         $S = S \backslash \{NA_i\}$
9:         $TSR_i = TSR_i - 1$
10:         $S = S \cup SelectTop\_K(P \backslash S, PST, 1)$
11:       **end if**
12:     **end for**
13: **end for**

---

## 5. Performance Evaluation

In this section, we conduct simulation experiments and analyze the experimental results. First, the experiment setup is introduced. Then, we compare the proposed multipath transmission scheme with other related works from different metrics. Finally, we analyze and discuss the experimental results.

### 5.1. Experiment Setup

We implement our proposed multipath transmission scheme based on NS-3 [46], which is designed to meet the needs of academic research and teaching. As shown in Figure 3, we use a real-world topology, the European Academic Network (GEANT2) [47], to comprehensively evaluate the performance of our multipath transport scheme. The default latency and bandwidth between backbone nodes in the topology are 100 Mbps and 1 ms, respectively. In addition, we added a data source node (S) and some receiver nodes (C1 to C5) at the edge of the network, marked with rectangles and triangles, respectively. The default bandwidth between the data source node (or receiver nodes) and the corresponding edge nodes is 1000 Mbps, and the default latency is 5 ms. In our scenario, chunk is the basic data unit, and the default size of chunk is set to 2 MB. A complete chunk will be divided into smaller segments for transmission, and the segment size is set to 1250 bytes. Additionally, each router has a maximum queue length of 1000 packets. To make a comprehensive

comparison, we use three transmission methods at the network layer, which are single path methods, i.e., Open Shortest Path First (OSPF) [48], ECMP, and our proposed multipath transmission scheme. OSPF, ECMP, and the proposed multipath transmission scheme belong to the category of network layer. At the transport layer, we use the transmission protocol proposed in previous work [35] to ensure reliable and stable data transmission.

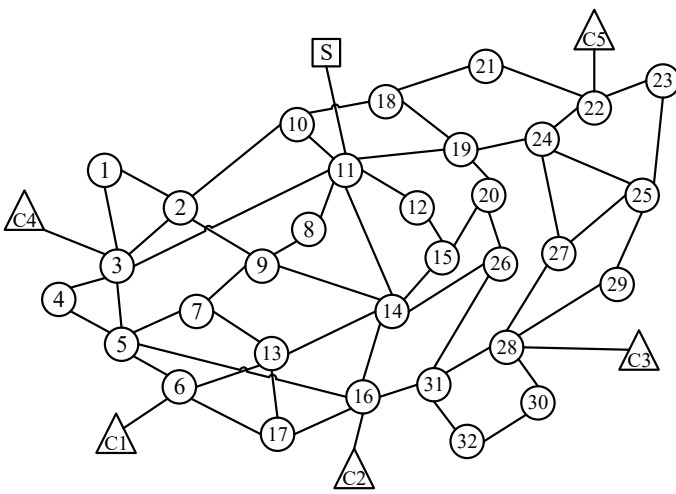

**Figure 3.** Simulation topology (GEANT2).

We evaluate the performance of our scheme based on three metrics: average throughput, average flow completion time (FCT), and average chunk completion time (CCT). The average CCT can reflect the response speed of each transmission method. The shorter the chunk completion time, the faster it can be delivered to the upper layer application. This is among the most important metrics for data transmission.

*5.2. Throughput*

Throughput is among the most important indicators of data transmission performance. In this subsection, we simulate the throughput of the multipath transmission scheme under different numbers of relay nodes and different link bandwidths.

Firstly, we set the number of relay nodes of GEANT2 to 5, 10, 15, 20, 25, and 30, respectively, to analyze the influence of the number of relay nodes on the throughput. The binding relationship between the MPSID and the NAs of these relay nodes has been registered on NRS. Then, we conducted five rounds of simulation. We let the data source node S continuously send data to a receiver node (C1 to C5), respectively, in each round and measured the steady-state throughput. As shown in Figure 4, it can be observed that when the number of relay nodes is ten, the average throughput can reach a maximum of about 169 Mbps. When the number of relay nodes is five, the average throughput is close to the maximum. This means that using only a few relay nodes can significantly improve the efficiency of multipath transmission. However, as the number of relay nodes increases, the average throughput decreases instead. The reason is that too many candidates increase the complexity of path selection and switching algorithms, so that it cannot quickly converge to the optimal relay. According to the experimental results, we set the number of relay nodes to ten in the following section.

Secondly, we simulated the average throughput under different bandwidths by changing the link bandwidth between the backbone nodes of the topology to 50 Mbps, 100 Mbps, 150 Mbps, and 200 Mbps, respectively. The experimental results are shown in Figure 5. It can be seen that the average throughputs of the single path method under different bandwidths were roughly 47 Mbps, 91 Mbps, 134 Mbps, and 179 Mbps, which is the lowest among the three methods. This is because the single path method cannot take advantage of other paths other than the default shortest path. It can also be observed that under different bandwidths, ECMP can improve the average throughput to about 62 Mbps, 113 Mbps,

158 Mbps, and 201 Mbps, respectively, but the improvement is not obvious. The reason is that ECMP is not always effective, since it only works on symmetric links. The symmetric links generally exist in data center networks [49], such as Fat-tree topology and Clos topology, but rarely exist in carrier networks or WANs. Compared with the other two methods, our proposed multipath transmission scheme can achieve the highest steady-state throughput under different link bandwidths, which are about 88 Mbps, 169 Mbps, 247 Mbps, and 302 Mbps, respectively. This also shows that our proposed method can achieve a significant improvement in average throughput.

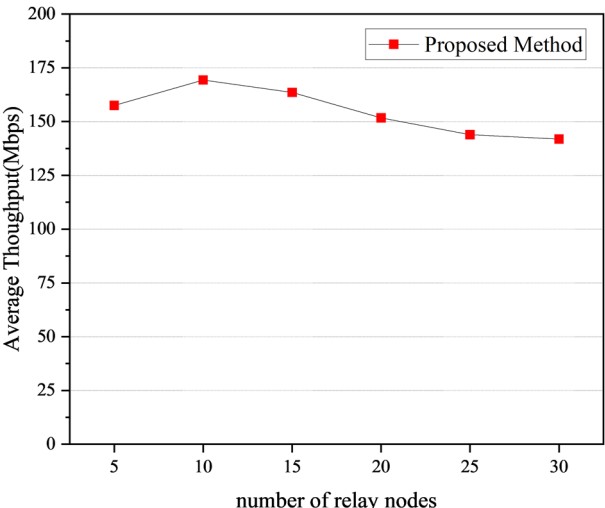

**Figure 4.** The average throughput under different number of relay nodes.

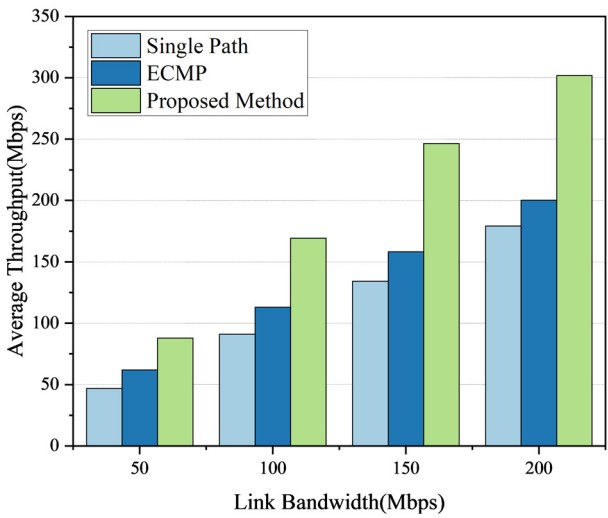

**Figure 5.** The average throughput under different link bandwidths.

### 5.3. Flow Completion Time

The average FCT can reflect the speed at which data transmission is completed, so we use this metric to evaluate our scheme. In this subsection, we simulated the average FCT of each transmission method under different link bandwidths by setting the link bandwidth between backbone nodes to 50 Mbps, 100 Mbps, 150 Mbps, and 200 Mbps, respectively. We performed five rounds of simulations. In each round, the data source node S connects with one of the receiver nodes (C1 to C5) and transmits 100 chunks. Figure 6 shows the average FCT of all connections transmitting 100 chunks under different bandwidths. As we expected, the average FCT of the single path method is the longest, about 36.5 s, 18.8 s, 12.8 s, 9.5 s, respectively, because it can only use the bandwidth resources on the shortest path. It

can also be observed that under different bandwidths, ECMP can reduce the average FCT to about 27.7 s, 15.2 s, 10.8 s, and 8.6 s, respectively. The data packets will be transmitted to the destination along multiple equal-cost paths, thus reducing the average FCT. Compared with the other two methods, our proposed method can significantly reduce the average FCT to 19.5 s, 10.1 s, 6.9 s, and 5.7 s, respectively. This illustrates the effectiveness of our proposed method.

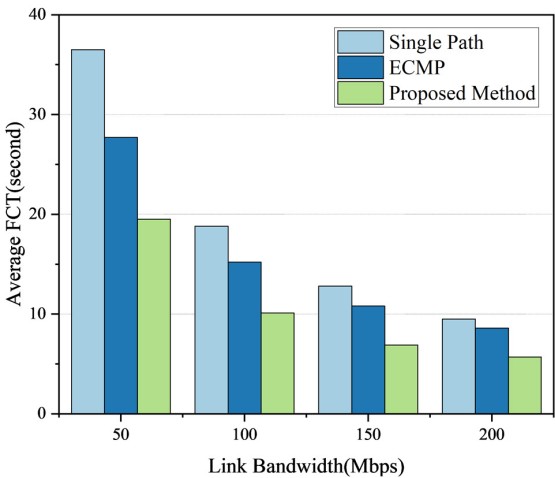

**Figure 6.** The average FCT under different link bandwidths.

### 5.4. Chunk Completion Time

In the ICN scenario, a chunk needs to be completely received before it can be delivered to the application layer. Therefore, the average chunk completion time is an important indicator showing the performance of the transmission method. The simulation steps are the same as those described in Section 5.3, and we measured the average chunk completion time under different bandwidths. Figure 7 shows the experimental results. It can be seen that as the bandwidth increases, the average CCT of each method decreases. The average CCT of the single path method is the highest, roughly 2.16 s, 1.75 s, 1.41 s, and 1.28 s under different bandwidths. It can also be observed that both ECMP and our proposed method can reduce the average CCT, and our proposed method is significantly more effective. Compared with the other two methods, our proposed method can reduce the average CCT to 1.44 s, 1.1 s, 0.74 s, and 0.56 s, respectively, under different bandwidths. This also shows that our proposed method can speed up the transmission, thus ensuring timely delivery of data to upper-layer applications.

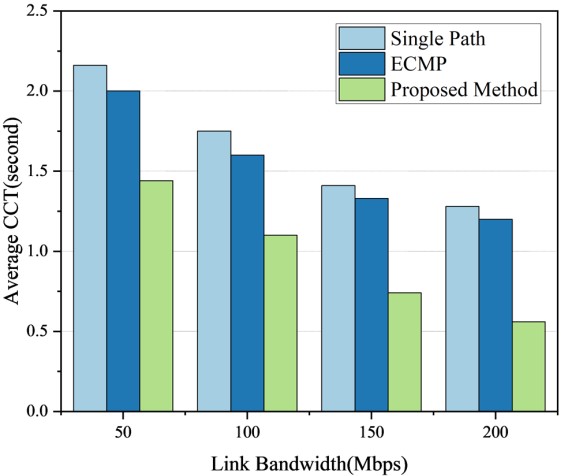

**Figure 7.** The average CCT under different link bandwidths.

In addition, as shown in Figure 8, we obtained the CCT cumulative distribution of data source node S and receiver node C1 when transmitting 100 chunks under 200 Mbps bandwidth. We can see that the proportions of CCT less than 1 s for the single path method, ECMP, and our proposed method are 0.74, 0.67, and 0.91, respectively, and the proportions of less than 2 s are 0.86, 0.98, and 0.98, respectively. Therefore, we can conclude that the single path method does not perform as well as the multipath method in reducing the CCT. Moreover, compared with ECMP, our proposed method can maintain the CCT at a lower level.

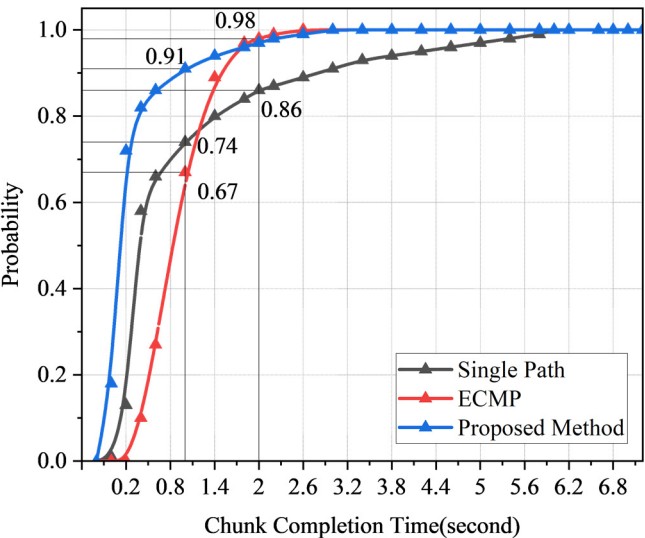

**Figure 8.** The cumulative distribution of the CCT.

## 6. Conclusions

In this paper, we propose a novel multipath transmission scheme which is applied in an IP-compatible ICN architecture. First, we introduce the overall layout of our ICN protocol stack and describe the principle and process of multipath transmission in detail. Then, we focus on how to make full use of multipath resources and propose a path management mechanism. In the path selection phase, we select several relay nodes as initial temporary solutions based on historical transmission information. In the path switching phase, path reselection can be performed during transmission by measuring the congestion degree of the selected sub-paths. Finally, we conduct simulation experiments to evaluate the performance of our proposed method. The experimental results show that our proposed method has an excellent performance in terms of average throughput, average flow completion time, and average chunk completion time.

In future research, we will focus on the following two aspects. Firstly, a reasonable data scheduling strategy is required to allocate the data ratio of each sub-path to ensure load balance. Secondly, the mathematical apparatus used in this paper is valid, but quite simple. Considering that the path selection method based on historical information cannot avoid the underlying overlapping paths, we will redesign the path selection strategy according to the network topology information obtained from the controller to eliminate the correlation of paths.

**Author Contributions:** Conceptualization, Y.X., H.N. and X.Z.; methodology, Y.X., H.N. and X.Z.; software, Y.X.; writing—original draft preparation, Y.X.; writing—review and editing, H.N. and X.Z.; supervision, X.Z.; project administration, X.Z.; funding acquisition, H.N. All authors have read and agreed to the published version of the manuscript.

**Funding:** This work was funded by the Strategic Leadership Project of the Chinese Academy of Sciences: SEANET Technology Standardization Research System Development (project no. XDC02070100).

**Institutional Review Board Statement:** Not applicable.

**Informed Consent Statement:** Not applicable.

**Data Availability Statement:** Not applicable.

**Acknowledgments:** We would like to express our gratitude to Jinlin Wang, Rui Han, and Zhiyuan Wang for their meaningful support of this work.

**Conflicts of Interest:** The authors declare no conflict of interest.

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
