# Peer review of "A Novel Multipath Transmission Scheme for Information-Centric Networking"

_futureinternet, doi:10.3390/fi15020080_

Round 1
Reviewer 1 Report
This article made a normal impression on me: an actual problem, good introduction, materials and methods, and also experimentation sections. Reference section is also adequate to the article.
Here are some comments that I have after reading this article:
1. The authors give a detailed review of the state of the problem, supported by many relevant references. I would additionally recommend (but do not insist) also to mention works from neighboring areas of science. For example self-organizing routing in wireless networks [https://doi.org/10.1145/1161089.1161093, https://doi.org/10.1109/INFCOM.2007.258]. Also, related problems (especially in the case of one-hop routing) are solved in the field of networks-on-chip (for example, fault-tolerant routing in [https://doi.org/10.1109/IECON48115.2021.9589829], or see a very good survey in the work [https://doi.org/10.1016/j.sysarc.2016.04.011].
2. Authors need to clearly describe the novelty of their solution, clearly explain why their approach is focused on ICN networks. What is its “information centricity”? Because the mathematical apparatus used is quite simple. It is not entirely clear how the proposed approach differs from other adaptive algorithms.
3. References to the NS3-simulator and GEANT2 topology are needed.
4. What is OSPF, ECMP? References are needed.
5. I did not find which traffic distribution scheme (traffic profile) was used in the simulation (and why).
6. There should be explanations with reference what "symmetric links" are.
I wish the authors great success in their work.
Reviewer 2 Report
This paper investigates a novel multipath transmission scheme applied in an IP-compatible ICN architecture to improve the data transmission efficiency and robustness.
1- Authors should clearly distinguish contributions from their previous work
Xu, Y.; Ni, H.; Zhu, X. An Effective Transmission Scheme Based on Early Congestion Detection for Information-Centric Network. Electronics 2021, 10, 2205.
2- Sections 3.1, 3.3., 5.1 and conclusions need to rewritten to eliminate similarity with Reference [33].
3- Complexity analysis should be given with respect to the other methods.
4- In Figures, instead of "Ours" use "Proposed Method"
